# The association of *EGFR* amplification with aberrant exon 20 insertion report using the cobas EGFR Mutation Test v2

Man-San Zhang[1], Yi-Chen Yeh[2,3], Hsien-Neng Huang[4,5], Long-Wei Lin[6], Yen-Lin Huang[7], Lei-Chi Wang[2,3], Lai-Jin Yao[1], Tze-Chun Hung[1], Yu-Fen Tseng[1], Yi-Hsuan Lee[1], Wei-Yu Liao[8], Jin-Yuan Shih[8], Min-Shu Hsieh[1,5,7] *

1 Department of Pathology, National Taiwan University Hospital, Taipei, Taiwan, 2 Department of Pathology and Laboratory Medicine, Taipei Veterans General Hospital, Taipei, Taiwan, 3 Institute of Clinical Medicine, National Yang Ming Chiao Tung University, Taipei, Taiwan, 4 Department of Pathology, National Taiwan University Hospital Hsin-Chu Branch, Hsinchu, Taiwan, 5 Graduate Institute of Pathology, National Taiwan University College of Medicine, Taipei, Taiwan, 6 Department of Pathology, National Taiwan University Hospital Yunlin Branch, Yunlin, Taiwan, 7 Department of Pathology, National Taiwan University Cancer Center, Taipei, Taiwan, 8 Division of Pulmonary and Critical Care Medicine, Department of Internal Medicine, National Taiwan University Hospital and National Taiwan University College of Medicine, Taipei, Taiwan

* mshsieh065@gmail.com

**Data Availability Statement:** All relevant data are within the manuscript and its Supporting Information files.

## Abstract

Determining the exact type of epidermal growth factor receptor (*EGFR*) exon 20 insertion (ex20ins) mutation in lung cancer has become important. We found that not all ex20ins mutations reported by cobas EGFR test v2 could be validated by Sanger sequencing even using surgical specimens with high tumor contents. This study aimed to validate the ex20ins results reported by the cobas test and to determine whether there were clinico-pathological factors associated with aberrant cobas ex20ins report. In total, 123 cobas-reported cases with ex20ins were retrospectively collected and validated by Sanger sequencing and Idylla assay. Clinicopathological features between ex20ins cobas +/Sanger+ group (n = 71) and cobas+/Sanger− group (n = 52) were compared. The Idylla assay detected ex20ins in 82.6% of cobas+/Sanger+ cases but only in 4.9% of cobas +/Sanger− cases. The cobas+/Sanger− group was significantly associated with higher tumor contents, poorly differentiated patterns, tumor necrosis, and a lower internal control cycle threshold value reported by the Idylla which suggesting the presence of increased *EGFR* gene copy numbers. *EGFR* fluorescence in situ hybridization (FISH) revealed the majority of cobas+/Sanger− group had *EGFR* high copy number gain (16%) or amplification (76%) according to the Colorado criteria. Among cases reported to have concomitant classic *EGFR* and ex20ins mutations by the cobas, the classic *EGFR* mutations were all detected by Sanger sequencing and Idylla, while the ex20ins mutations were undetected by Sanger sequencing (0%) or rarely reported by Idylla assay (3%). FISH revealed high *EGFR* copy number gain (17.9%) and amplification (79.5%) in cases reported having concomitant classic *EGFR* and ex20ins mutations by the cobas. This study demonstrated an unusually high frequency of *EGFR* amplification in cases with aberrant cobas ex20ins report which could not be validated by Sanger sequencing or Idylla assay. Ex20ins

**Funding:** Min-Shu Hsieh received funding supported by National Science and Technology Council, Taiwan (112-2320-B-002-047) and National Taiwan University Hospital, Taipei, Taiwan (grant numbers NTUH112-N0023 and NTUH112-S0046). The funders had no role in study design, data collection and analysis, decision to publish, or preparation of the manuscript.

**Competing interests:** The authors have declared that no competing interests exist.

reported by the cobas test should be validated using other methods especially those reported having concomitant ex20ins and classic *EGFR* mutations.

## Introduction

Lung cancer is the leading cause of cancer-related death in Taiwan, and more than 50% of newly diagnosed cases are stage III or IV [1]. In Taiwan, epidermal growth factor receptor (*EGFR*) mutations are the most common disease-associated mutations in lung cancer with an incidence rate around 55% [2, 3], and several *EGFR* tyrosine kinase inhibitors (EGFR-TKIs) are used as first-line therapy [3–7]. Lung cancers with classic *EGFR* mutations, such as exon 19 deletion (ex19del) and the exon 21 p.Leu858Arg (L858R) point mutation, are responsive to first-to-third-generation EGFR-TKIs, including gefitinib, erlotinib, afatinib, and osimertinib [2–5]. Based on the structural changes and drug sensitivity, *EGFR* mutations can be separated into four distinct subgroups as classical-like, T790M-like, ex20 loop insertion, and P-loop αC-helix compression types [8].

Patients with uncommon *EGFR* mutations, such as p.G719X, p.S768I, or p.L861Q, have been shown to be less responsive to EGFR-TKIs, and patients with exon 20 insertion (ex20ins) may be non-responsive to traditional EGFR-TKIs [2–9]. *EGFR* ex20ins mutations are genetically heterogeneous and comprise approximately 10% of all *EGFR* mutations in Western countries and 2–4% of activating *EGFR* mutations in East Asia (Taiwan, Japan, China, and South Korea) [2, 10–18]. More than 50 different types of ex20ins mutations have been identified, and these genetically heterogeneous mutations occur in the helical (positions 761–766), near-loop (positions 767–772), and far-loop (positions 773–774) regions [10–16]. The exact type of ex20ins mutation is clinically important as patients with ex20ins mutations in the near-loop region (positions 767–772) are resistant to conventional EGFR-TKIs while those with ex20ins mutations in the helical region (positions 761–766) are responsive to first-generation EGFR-TKIs [7, 9, 11]. With the development of several ex20ins-selective drugs such as mobocertinib, amivantamab, poziotinib, and furmonertinib, it is clinically important to determine the exact ex20ins mutation type [8–19].

Several molecular testing methods have been used to detect druggable EGFR mutations. Sanger sequencing has traditionally been considered as the gold standard method but with a low sensitivity which typically requires 20% of mutated alleles [20]. Real-time polymerase chain reaction (RT-PCR)-based tests have been developed and widely used in Taiwan. The Roche cobas EGFR Mutation Test v2 (Roche Molecular Diagnostics, Pleasanton, CA, USA), an RT-PCR test approved by the U.S. Food and Drug Administration as a companion diagnostic tool for EGFR-TKI therapy for advanced non-small cell lung cancer, is the most popular testing platform in Taiwan. This test can detect 42 specific mutations in exons 18 to 21 of the human *EGFR* gene with a detection limit of ~5% of mutant alleles. For ex20ins mutations, the cobas test covers the five most common mutations that lead to four types of amino acid changes: p.V769_D770insASV, p.D770_N771insSVD, p.D770_N771insG, and p.H773_V774insH. The Idylla™ EGFR Mutation Test (Biocartis, Mechelen, Belgium) is a fully automated RT-PCR based molecular diagnostic kit for the qualitative detection of 51 *EGFR* mutations (including five ex20ins mutations identical to those designed in the cobas test) for FFPE tissue samples with a limit of detection ~2–5% of variant allele frequency [21, 22]. Next-generation sequencing (NGS) is more sensitive than RT-PCR based tests with a limit of detection around 1% of variant allele frequency [23].

Since *EGFR* ex20ins were considered drug-resistant mutations at the time when the cobas was developed, the cobas test has some limitations in the detection of ex20ins. First, the cobas test reports different ex20ins variants together as "ex20ins mutation detected" without differentiating specific mutation types. Second, lung cancer with ex20ins mutation in the helical region (p.A763_Y764insFQEA) cannot be assessed by the cobas test, and patients with helical region mutations are responsive to traditional EGFR-TKIs [17]. Third, more than 50 different types of ex20ins mutations have been identified, and the cobas test could only detect approximately 40% of all ex20ins-positive cases [15–19].

To determine *EGFR* mutations, Sanger sequencing was performed at the National Taiwan University Hospital (NTUH) between January 02, 2014, and January 19, 2019, after which it was replaced by the cobas EGFR Mutation Test v2 (Roche Diagnostics, Tucson, AZ). In our hospital all ex20ins-positive cases reported by the cobas will be sent for Sanger sequencing under the request by clinical oncologists. Although greater sensitivity of the cobas method has been described, ex20ins mutations in many cases (approximately one-third of ex20ins cases reported by cobas) could not be confirmed by Sanger sequencing even using surgical specimens with high tumor contents.

Since there was disparity in ex20ins results between cobas and other testing methods, this study aimed to validate the ex20ins results reported by the cobas test and to determine whether there were clinicopathological factors associated with aberrant cobas ex20ins report.

## Materials and methods

### Case selection and design of the study

Based on the clinical observation that cobas test might give aberrant *EGFR* ex20ins reports, a multicenter, cross-sectional study was designed to compare the ex20ins results among different testing methods. The cases of 5,108 patients who underwent *EGFR* mutation testing in our department between January 2015 and December 2022 were reviewed. Among these, 29 ex20ins-positive cases from 1,019 tested by Sanger sequencing (January 02, 2015, to January 19, 2019) and 113 ex20ins-positive cases from 4,089 cases (including 30 cases with concomitant classic *EGFR* mutations) reported by the cobas EGFR Mutation Test v2 (January 20, 2019, to December 31, 2022) were identified at the NTUH, National Taiwan University Cancer Center, NTUH Hsin-Chu Branch, and NTUH Yunlin Branch. An additional 10 cases reported to have concomitant classic *EGFR* mut+/ex20ins+ by cobas testing were retrieved from the Taipei Veterans General Hospital. The first aim was to compare tumor contents between cobas +/Sanger+ and cobas+/Sanger− ex20ins groups to determine whether the different testing sensitivities of cobas and Sanger sequencing leading to the disparity of ex20ins results. Clinicopathological features including tumor contents were compared between cobas+/Sanger+ and cobas+/Sanger− ex20ins groups. Then Idyalla EGFR Assay was used to validate the ex20ins results of both groups, and the IC Ct values from two groups were compared.

The histologic grading was reviewed by two senior thoracic pathologists (MSH and YHL) and categorized according to the recommendations of the International Association for the Study of Lung Cancer as grade 1 (well-differentiated): lepidic-predominant with no or <20% high-grade pattern; grade 2 (moderately differentiated): acinar or papillary-predominant with no or <20% high-grade pattern; grade 3 (poorly differentiated): tumor with ≥20% high-grade pattern (solid, micropapillary, cribriform, or complex glandular pattern) [24]. Twenty patients underwent liquid NGS, and their results were retrieved from their medical records. This study was approved by the Research Ethics Committee of the National Taiwan University Hospital (202206016RIND), and data had been accessed from July 10, 2022 to May 15, 2023.

## Mutation analysis

For the cobas EGFR Mutation Test, DNA was isolated from formalin-fixed paraffin-embedded specimens using the cobas DNA Sample Preparation Kit (Roche Molecular Systems, Inc., Tucson, AZ). The extracted DNA was subjected to real-time polymerase chain reaction (PCR)-based *EGFR* mutation tests using the cobas EGFR Mutation Test v2 (Roche Molecular Systems, Inc.), as per the manufacturer's instructions.

For Sanger sequencing, DNA was extracted from formalin-fixed paraffin-embedded specimens using a DNA Extraction Kit (Qiagen, Hilden, Germany) and subjected to PCR (Biometra Thermocycler TProfessional Basic 96) using specific primers (See S1 Table in S1 File). Subsequently, the PCR amplicons were purified using ExoSAP-IT (Thermo Fisher Scientific, Waltham, MA). Purified DNA was cycle-sequenced using ABI BigDye V3.1 and electrophoresed using an ABI 3730xl genetic analyzer (Applied Biosystems, Waltham, MA). The sequences were compared with the GenBank-archived human *EGFR* sequences.

For Idylla EGFR Mutation Testing, 5 μm formalin-fixed paraffin-embedded tissue sections were placed into the Idylla EGFR Mutation test cartridge and submitted to the fully-automated Idylla platform (Biocartis). After 150 min, the test was automatically performed inside the cartridge, and the final report was released directly from the system after automatic onboard post-PCR curve analysis. IC Ct values between ex20ins cobas+/Sanger+ and cobas+/Sanger− groups were compared.

## *EGFR* FISH

The lower IC Ct value in the cobas+/Sanger- ex20ins group raised the suspicion that high *EGFR* copy numbers may be associated with aberrant cobas ex20ins report. FISH was used to compare *EGFR* gene copy numbers between x20ins cobas+/Sanger+ and cobas+/Sanger− groups using the Colorado Scoring Criteria [25]. Fig 1 shows the algorithm for groups cases for *EGFR* FISH.

Using the ZytoLight SPEC EGFR/CEN 7 dual-color probe (ZytoVision GmbH, Bremerhaven, Germany), *EGFR* FISH was performed on 104 cases (54 cobas+/Sanger+ cases and 50 cobas+/Sanger− cases) with available specimens. Physically linked doublet or triplet *EGFR* signals were considered single signals. In each case, 50 non-overlapping tumor cell nuclei were evaluated (by MSZ and MSH) according to the Colorado Scoring Criteria: disomy (score = 1), low trisomy (score = 2), high trisomy (score = 3), low polysomy (score = 4), high polysomy (score = 5, defined as ≥40% of cells displaying ≥4 copies of the EGFR signal), and gene amplification (score = 6, defined as (a) an EGFR to CEP7 ratio ≥2 over all scored nuclei and calculated using the sum of EGFR divided by the sum of CEP7 when mean CEP7 per cell is ≥2 copies, or (b) the presence of gene cluster (≥4 spots) in ≥10% of tumor cells, or (c) at least 15 copies of the EGFR signals in ≥10% of tumor cells.) [25]. Gene amplification was further categorized as (A) large clusters of *EGFR* signal with an *EGFR* to CEP7 ratio >2; (B) small clusters of *EGFR* signal (≥4 signals) with an *EGFR* to CEP7 ratio >2; (C) co-localized clusters of *EGFR* and CEP7 signals; (D) very high number of balanced (*EGFR* ≥15) and CEP7 signals; (E) *EGFR* signals as numerous double minutes (≥15 copies) according to the "updated Colorado score [25]" for scoring and interpreting clusters (Fig 2).

## Statistical analysis

Continuous variables between groups were analyzed by Mann–Whitney U test, and those among the three groups (Colorado score 6, 5, and 1–4) were analyzed using one-way ANOVA test, with *p*-values < 0.05 considered statistically significant. Categorical variables were analyzed using the two-sided Fisher's exact test. Cohen's kappa was used to compare the results of

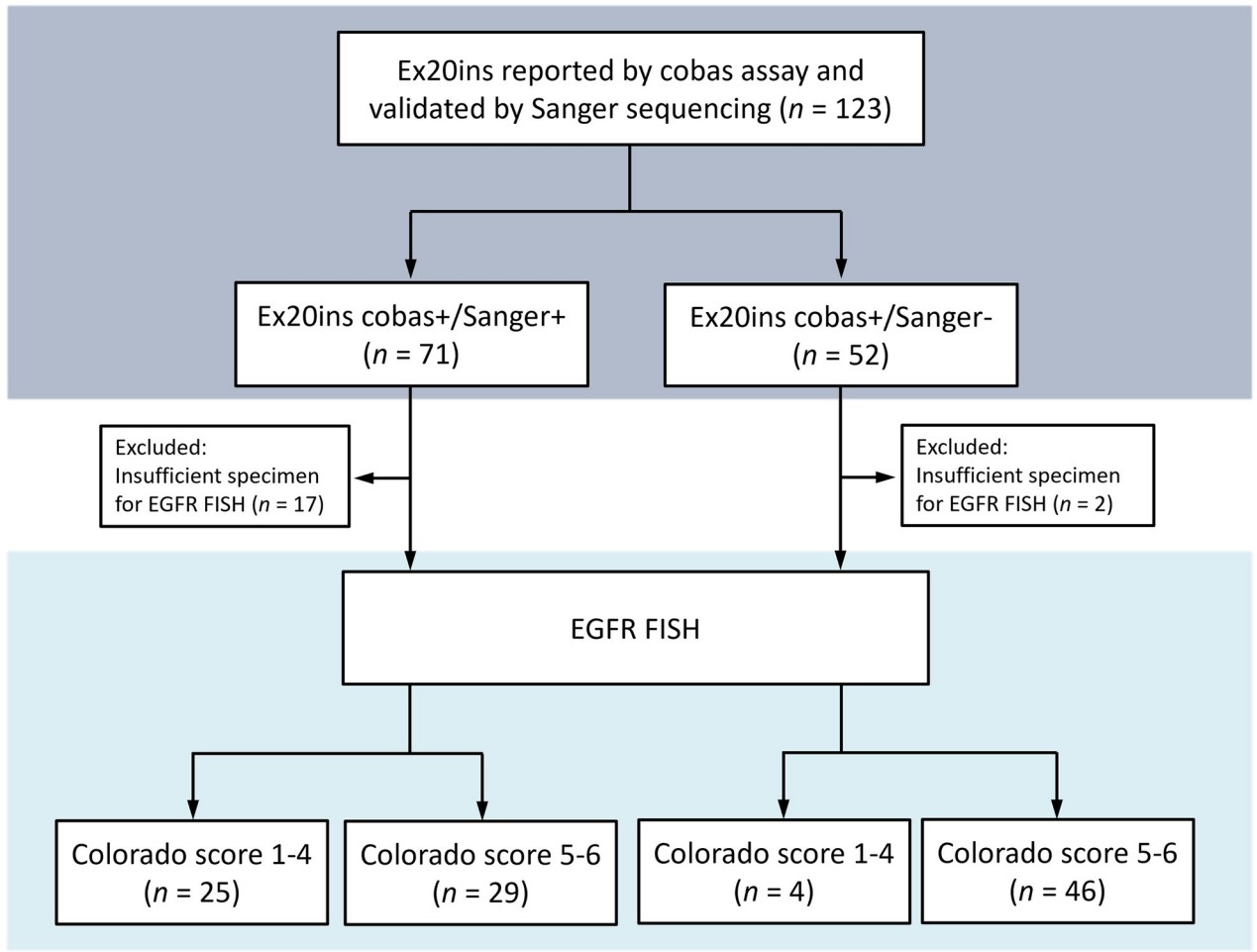

**Fig 1. Algorithm for grouping cases for *EGFR* FISH.**

the Idylla EGFR test and Sanger sequencing. The STATA software (version 13.0; StataCorp LLC, College Station, TX, USA) was used for all the statistical analyses.

## Results

### The emergence of cases reported to have concomitant ex20ins and classic EGFR mutations after the use of the cobas test and discrepancies with Sanger sequencing

Table 1 summarizes the cases using two different *EGFR* mutation testing methods (Sanger sequencing and cobas EGFR test v2) in our department from January 2015 to December 2022. The *EGFR* mutation rate was higher after the use of cobas testing ($p<0.001$) while the ex20ins rates were equal (2.8%) for both testing methods. However, more than one-third of the ex20ins-positive cases (37.2%) reported by cobas testing did not have their ex20ins mutations detected by Sanger sequencing. Moreover, 30 patients reportedly had concomitant classic *EGFR* mut+/ex20ins+ using the cobas test; however, none of their ex20ins could be confirmed by Sanger sequencing. Concomitant classic *EGFR* mut+/ex20ins+ were not observed during the Sanger sequencing period (January 2015 to January 2019) ($p<0.001$).

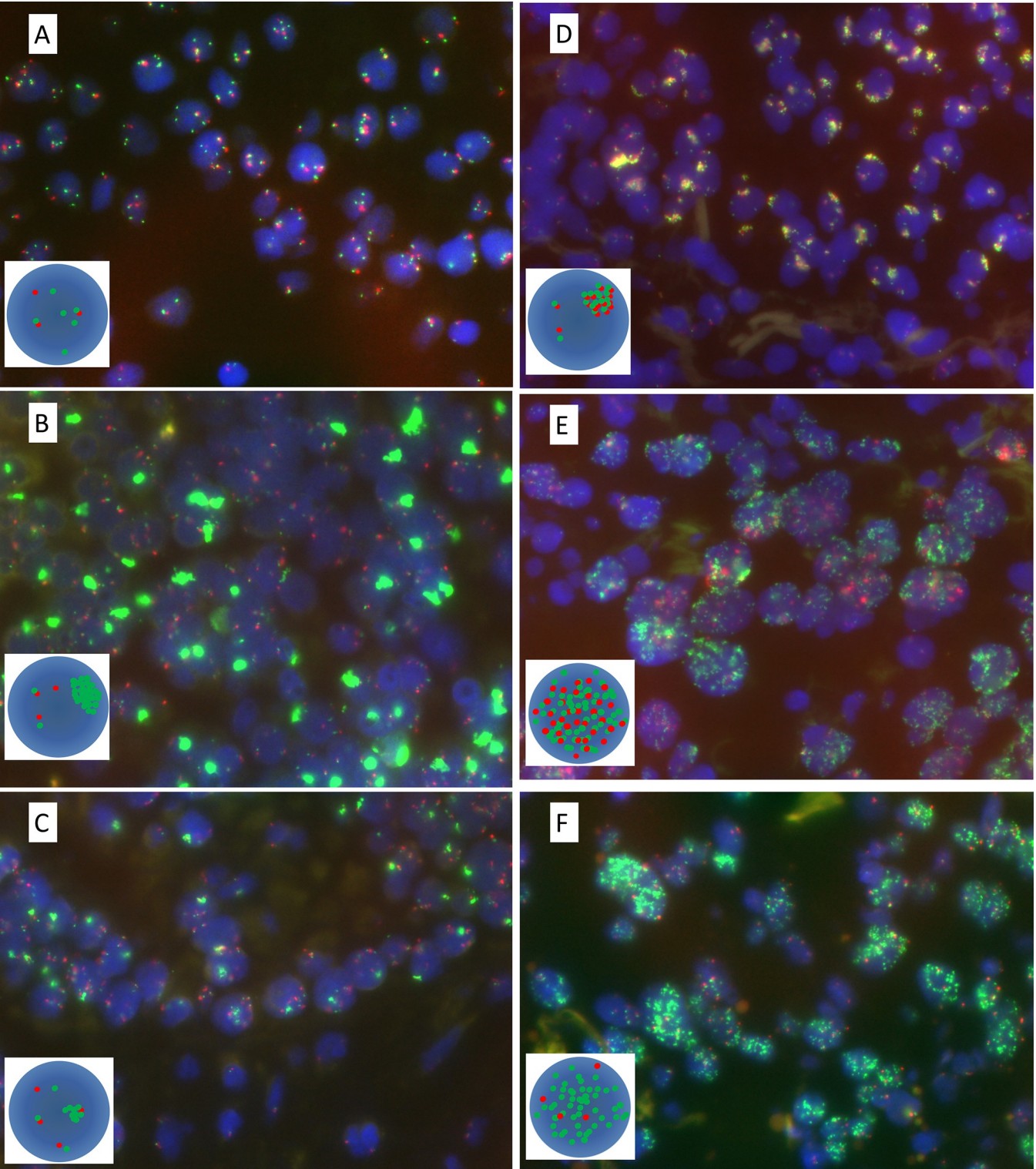

**Fig 2. *EGFR* FISH high copy number and amplification patterns.** (A) Colorado score 5: high copy number. (B–F), Colorado score 6: different types of *EGFR* amplification. (B): Large clusters of *EGFR* signals; (C): Small clusters of *EGFR* signals; (D): Co-localized clusters of *EGFR* and CEP7 signals; (E): Very high number of balanced green (*EGFR*) and red (CEN7) signals; (F): Double minutes. FISH, fluorescence in situ hybridization.

**Table 1. Comparison of *EGFR* mutation results using Sanger sequencing and the cobas EGFR assay at NTUH between January 2015 and December 2022.**

|  | Sanger sequencing (%) | cobas EGFR assay (%) | *p*-value | Total (%) |
|---|---|---|---|---|
| *EGFR* tests | 1,019 | 4,089 | <0.001 | 5,108 |
| EGFR mut+ | 672 (65.9) | 2,997 (73.3) |  | 3,669 (71.8) |
| EGFR mut− | 347 (34.1) | 1,092 (26.7) |  |  |
| *EGFR* mutation types |  |  | 0.281 |  |
| ex19del | 294 (28.9) | 1,056 (25.8) |  | 1,350 (26.4) |
| ex21p.L858R | 310 (30.4) | 1,288 (31.5) |  | 1,598 (31.3) |
| ex20ins | 29 (2.8) | 113 (2.8) |  | 142 (2.8) |
| ex20ins case | 29 | 113 |  | 142 |
| Sanger sequencing (+) | 29 (100) | 71 (62.8) |  | 100 (0.4) |
| Helical region (positions 761–766) | 4 (13.8) | 0 (0) | 0.005 | 4 (4.4) |
| Near-loop region (positions 767–772) | 21(72.4) | 63 (88.7) |  | 84 (93.3) |
| Far-loop region (positions 773–775) | 4 (13.8) | 8 (11.3) |  | 12 (13.3) |
| Sanger sequencing (−) | 0 (0) | 42 (37.2) | <0.001 | 42 (29.6) |
| Concomitant ex20ins and classic *EGFR* mutations | 0 (0) | 30* (26.5) | <0.001 | 30 (21.1) |

*All cases in which ex20ins mutations were not detected using Sanger sequencing.

Abbreviations: *EGFR*, epidermal growth factor receptor; ex20ins, exon 20 insertion; mut, mutation; ex19del, exon 19 deletion; NTUH, National Taiwan University Hospital

## The cobas+/Sanger− ex20ins group had a higher tumor content, a higher histologic grade, more frequent tumor necrosis, and a lower IC Ct value

Pathologically, cobas-identified ex20ins-positive cases were further divided into two groups: cobas+/Sanger+ group (ex20ins mutations detectable by Sanger sequencing, n = 71) and cobas+/Sanger− group (ex20ins mutations undetectable by Sanger sequencing, n = 52 [42 from NTUH and 10 from Taipei Veterans General Hospital]). Of these cases, 104 cases had available biopsy or resection specimens for the evaluation of histological grading and tumor necrosis. The clinicopathological features of the patients in these groups are listed in the S2 File and summarized in Table 2. There were no significant differences in age or sex between the two

**Table 2. Clinicopathological features between ex20ins cobas+/Sanger+ and cobas+/Sanger-groups.**

| EGFR ex20ins | cobas+/Sanger+ | cobas+/Sanger− | *p*-value |
|---|---|---|---|
| Case number | 71 | 52# |  |
| Sex (male:female) | 24:47 | 23:29 | 0.264 |
| Age (years), range (mean) | 36–86 (62) | 42–89 (66.5) | 0.0694 |
| Tumor content %, range (mean) | 5–80 (25.6) | 10–90 (45.5) | <0.001 |
| Histologic grading, n (%) |  |  | 0.01 |
| 1 (lepidic) | 2 (3.7) | 1 (2) |  |
| 2 (acinar/papillary) | 19 (35.2) | 5 (10) |  |
| 3 (micropapillary, solid, complex glandular structures) | 33 (61.1) | 44 (88) |  |
| Necrosis, n (%) | 10 (18.5) | 28 (56) | <0.001 |
| Idylla EGFR assay, n | 46 | 41 |  |
| Idylla ex20ins(+), n (%) | 38 (82.6) | 2 (4.9) |  |
| Idylla ex20ins(−), n (%) | 8 (17.4) | 39 (95.1) |  |
| IC Ct value, range (mean) | 15.7–28.4 (21.2) | 14–28.5 (18.6) | <0.001 |

# 42 cases from NTUH and 10 cases from TVGH

Abbreviations: Ct, cycle threshold; *EGFR*, epidermal growth factor receptor; ex20ins, exon 20 insertion; IC, internal control; NTUH, National Taiwan University Hospital; TVGH, Taipei Veterans General Hospital

groups. The cobas+/Sanger− group had a higher tumor content ($p<0.001$), more poorly differentiated (grade 3) tumors ($p = 0.01$), and more frequent tumor necrosis ($p<0.001$).

A total of 87 cases had available specimens for Idyalla EGFR test. Ex20ins mutations were reported in 38 of 46 (82.6%) cobas+/Sanger+ cases and two of 41 (4.9%) of cobas+/Sanger- cases ($p<0.001$). Unlike cobas test, the internal control (IC) cycle threshold (Ct) value of Idylla was available. The IC Ct value was also significantly lower in the cobas+/Sanger− group ($p<0.001$), suggestive of increased *EGFR* gene copy numbers in this group. In this study, Sanger sequencing showed a much higher agreement with the Idylla (77/87, 88.5%) than the cobas (71/123, 57.7%) test (Table 2). Ten cases with inconsistent *EGFR* results between Sanger sequencing and Idylla testing are summarized in S2 Table in S1 File. Of the nine ex20ins + cases not detected by Idylla, the ex20ins mutations in seven were not included in the design of the Idylla assay. This finding also reflects that the cobas had cross-reactivity in the detection of ex20ins because these mutations were not included in the design of the cobas test. After excluding the seven cases whose ex20ins mutations were not included in the detection range of the Idylla assay, the agreement between Sanger sequencing and Idylla testing increased to 96.3% (77/80).

## Significant disparity in rates of EGFR amplification between cobas+/Sanger − and cobas+/Sanger+ groups

*EGFR* FISH revealed that a significantly greater number of cases with high *EGFR* copy numbers (Colorado score 5 and 6) (92%) in the cobas+/Sanger− group ($p<0.001$) (Table 3). *EGFR* amplification (Colorado score 6) represented the most important disparity between the cobas +/Sanger− group (76%) and the cobas+/Sanger+ group (14.8%) (Fig 3). The most common amplification pattern was large clusters of *EGFR* signals with an *EGFR* to CEP7 ratio >2 (pattern A), followed by *EGFR* signals with numerous double minutes (≥15 copies) (pattern E).

## The majority of cases reported having concomitant classic *EGFR* mut +/ex20ins+ by the cobas test had *EGFR* amplification

In our department, cases reported as having concomitant classic *EGFR* mut+/ex20ins+ have appeared since the initiation of the cobas test in January 2019. With additional 10 cases from

**Table 3. Results of *EGFR* FISH and Idylla assay between ex20ins cobas+/Sanger+ and cobas+/Sanger-groups.**

| EGFR ex20ins | cobas+/Sanger+ | cobas+/Sanger− | *p*-value |
|---|---|---|---|
| *EGFR* FISH Colorado score, n (%) | 54 | 50[#] | <0.001 |
| Score 1–4 | 25 (46.3) | 4 (8) | |
| Score 5 | 21 (38.9) | 8 (16) | |
| Score 6 | 8 (14.8) | 38 (76) | |
| Score 6 patterns (n) | | | |
| A | 2 | 24 | |
| B | 1 | 2 | |
| C | 1 | 2 | |
| D | 1 | 1 | |
| E | 3 | 9 | |

[#] 40 cases from NTUH and 10 cases from TVGH

Abbreviations: *EGFR*, epidermal growth factor receptor; ex20ins, exon 20 insertion; FISH, fluorescence in situ hybridization; IC CT, internal control cycle threshold; NTUH, National Taiwan University Hospital; TVGH, Taipei Veterans General Hospital

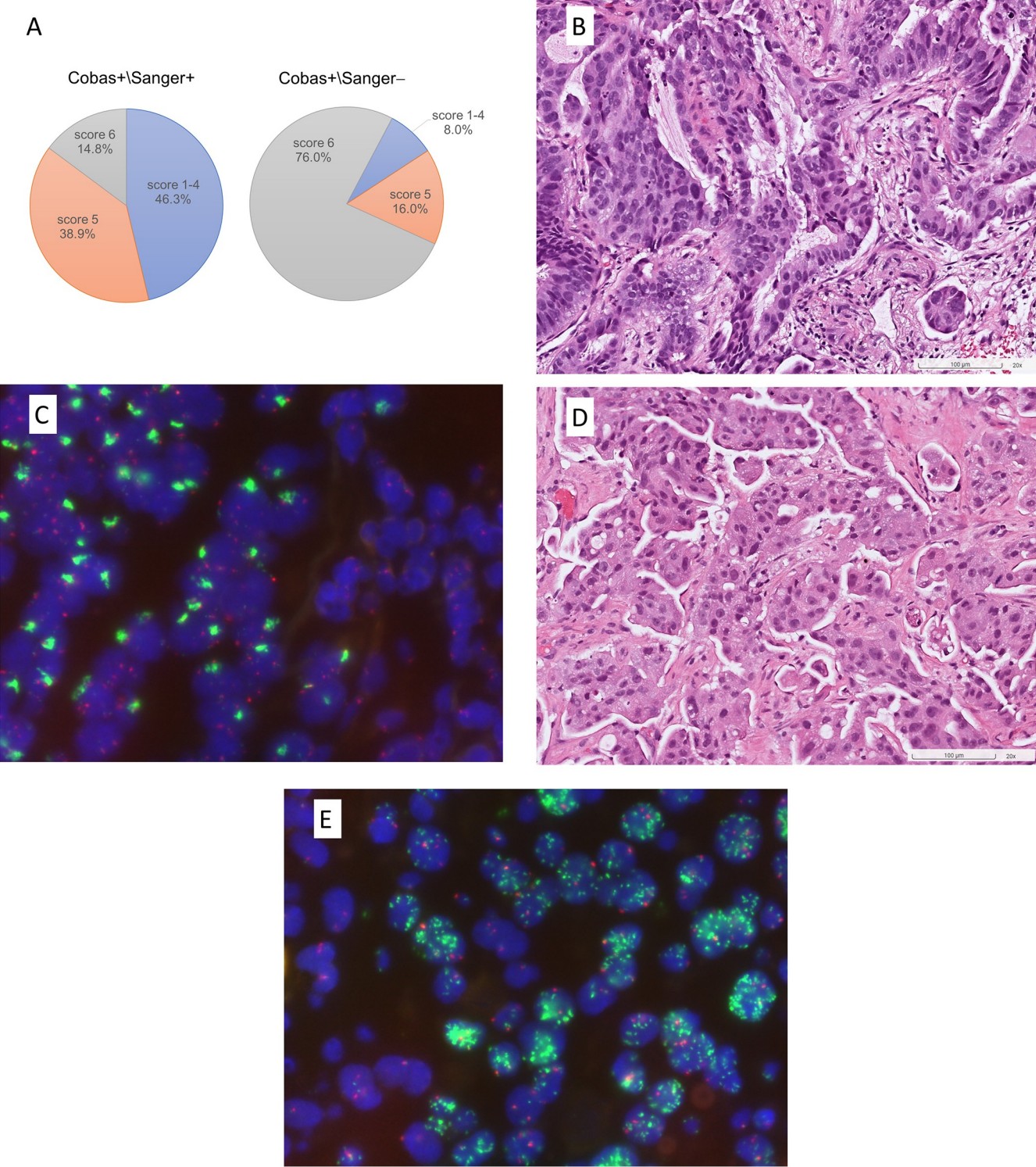

**Fig 3. *EGFR* FISH between cobas+/Sanger+ and cobas+/Sanger− group.** (A) The percentage of *EGFR* high copy number gain (Colorado score 5) + amplification (Colorado score 6) was significantly higher in the cobas+/Sanger− group compared with the cobas+/Sanger+ group. (B) This case reported to have ex20ins mutations by cobas test but not detected by Sanger sequencing and Idylla assay, with high-grade complex glandular structures and (C) *EGFR* amplification featured by large clusters of *EGFR* signals (green). (D) This case reported to have ex20ins mutations by cobas test but not detected by Sanger sequencing and Idylla assay, with high-grade micropapillary patterns and (E) *EGFR* amplification (more than 15 copies per cell). Hematoxylin and eosin (B, D) and *EGFR* FISH, green (*EGFR*) and red (CEN7) signals (C, E).

Taipei Veterans General Hospital, a total of 40 cases reported to have concomitant classic *EGFR* and ex20ins were collected and their clinicopathological features are summarized in Table 4. Cases with concomitant classic *EGFR* mut+/ex20ins+ had a high tumor content (mean 49.4%), 90% of them were poorly differentiated (grade 3), and more than half had tumor necrosis (57.5%). Thirty-nine cases had specimens available for *EGFR* FISH, and nearly all cases (38/39) had a high *EGFR* copy number gain either as high polysomy (Colorado score 5, 17.9%) or amplification (Colorado score 6, 79.5%) (Fig 4).

Among cobas-reported classic *EGFR* mut+/ex20ins+ cases, Sanger sequencing and Idylla test were successfully performed in 39 and 33 cases, respectively. The classic EGFR mutations

**Table 4. Clinicopathological features of cases reported having concomitant classic *EGFR* and ex20ins mutations by the cobas test.**

| cobas EGFR assay | ex20ins+/classic *EGFR* mut+ |
|---|---|
| Case number | 40[#] |
| Sex (male:female) | 14:26 |
| Age (years), range (mean) | 45–89 (67.4) |
| Tumor content %, range (mean) | 10–90 (49.4) |
| Histologic grading, n (%) | |
| 1 (lepidic) | 0 (0) |
| 2 (acinar/papillary) | 4 (10) |
| 3 (micropapillary, solid, complex glandular structures) | 36 (90) |
| Necrosis, n (%) | 23 (57.5) |
| *EGFR* FISH Colorado score, n (%) | 39 |
| Score 1–4 | 1 (2.6) |
| Score 5 | 7 (17.9) |
| Score 6 | 31 (79.5) |
| Score 6 patterns | |
| A | 20 |
| B | 1 |
| C | 2 |
| D | 1 |
| E | 7 |
| Sanger sequencing, n (%) | 39 |
| ex20ins+/classic *EGFR* mut− | 0 |
| ex20ins−/classic *EGFR* mut− | 0 |
| ex20ins+/classic *EGFR* mut+ | 0 |
| ex20ins−/classic *EGFR* mut+ | 39 (100) |
| Idylla EGFR assay, n (%) | 33 |
| ex20ins+/classic *EGFR* mut− | 0 |
| ex20ins−/classic *EGFR* mut− | 0 |
| ex20ins+/classic *EGFR* mut+ | 1 (3.0) |
| ex20ins−/classic *EGFR* mut+ | 32 (97.0) |
| IC Ct value, range (mean) | 14.8–23.8 (18.6) |

[#] 30 patients from NTUH and 10 patients from TVGH. ex20ins mutations were not detected using Sanger sequencing in any of the 40 patients.

Abbreviations: *EGFR*, epidermal growth factor receptor; ex20ins, exon 20 insertion; FISH, fluorescence in situ hybridization; mut, mutation; IC Ct, internal control cycle threshold; NTUH, National Taiwan University Hospital; TVGH, Taipei Veterans General Hospital

of all tested cases were all detected by Sanger and Idylla, while the ex20ins was not detected by Sanger sequencing (0/39) and only one (1/33, 3%) reported by Idylla (Table 4).

## Comparison of cobas testing, Sanger sequencing, *EGFR* FISH, and liquid NGS results

Twenty cases reported to have ex20ins by the cobas test (9 cobas+/Sanger+ and 11 cobas +/Sanger− cases) underwent liquid NGS, and the results are summarized in Table 5. Liquid NGS successfully identified ex20ins mutations in 7 of 9 cobas+/Sanger+ ex20ins cases, but only in 1 of 11 cobas+/Sanger− ex20ins cases ($p$ = 0.005). Among the six cases reported to have concomitant classic *EGFR* mut+/ex20ins+ by cobas testing, liquid NGS identified all classic *EGFR* mutations, *EGFR* amplification in four cases, and no ex20ins mutations. Among these 20 cases of *EGFR* mutations, liquid NGS showed higher agreement with Sanger sequencing (17/20, 85%) than with the cobas test (8/20, 40%).

## Discussion

This study reviewed cases reported having *EGFR* ex20ins mutations by the cobas test and compared the *EGFR* mutation results among different testing methods. In our hospital system, more than one-third of the ex20ins mutation reported by cobas testing could not be validated by Sanger sequencing or Idylla assay. Sanger sequencing is less sensitive than RT-PCR tests like cobas or Idylla. The limit of detection is approximately 15–20% for mutant allele frequency by Sanger sequencing, and 5–10% by the cobas test or the Idylla quantitative PCR (qPCR)-based method [20–23, 26]. Some may argue that the cobas+/Sanger− ex20ins cases reflect the different limits of detection between these two methods. Surprisingly, this study revealed that tumor content was significantly higher in the cobas+/Sanger− group (10–90%, mean 45.5%) compared with that in the cobas+/Sanger+ group (5–80%, mean 25.6%). Therefore, the limit of detection theory cannot explain why Sanger sequencing could not detect ex20ins mutations in more than one-third of the cobas-identified ex20ins-positive cases (37.2%), even when using surgical specimens with high tumor content. The sensitivity of the Idylla test is comparable to that of the cobas test. However, the Idylla assay detected ex20ins in 82.6% of cobas +/Sanger+ cases but only in 4.9% of cobas+/Sanger− cases. The Idylla assay showed poor agreement with the cobas test (46%, 40/87) compared to Sanger sequencing (77/87, 88.5%) in the detection of ex20ins mutations. The agreement between Sanger sequencing and Idylla was even higher (77/80, 96.3%) after excluding seven cases whose ex20ins mutations were not included in the detection range of the Idylla assay. Since both the cobas test and Idylla EGFR assay are qPCR-based platforms with similar sensitivities, this study revealed that the cobas test may yield aberrant ex20ins reports that cannot be validated by the Idylla test or Sanger sequencing.

The Idylla assay also showed the cobas+/Sanger- ex20ins group had a lower IC Ct value compared with the cobas+/Sanger+ ex20ins group. Since a low IC Ct value reported by the RT-PCR platform implies the presence of increased gene copy numbers in the tested specimen, *EGFR* FISH was then used to compare the *EGFR* gene copy numbers between cobas+/Sanger + and cobas+/Sanger− groups. Our study found an unusually high frequency of *EGFR* amplification in cobas+/Sanger− groups, raising the possibility that high *EGFR* copy numbers, especially *EGFR* amplification, may lead to aberrant ex20ins reports from the cobas test. The prevalence of *EGFR* amplification in non-small cell lung cancer ranges from 6–19% [27–32]. In this study, similar *EGFR* amplification rates were observed in the cobas+/Sanger+ ex20ins group (14.8%). However, it increased to 76% in the cobas+/Sanger− ex20ins group and 79.5% in cases with concomitant classic *EGFR* mut+/ex20ins+ reported by the cobas test. The

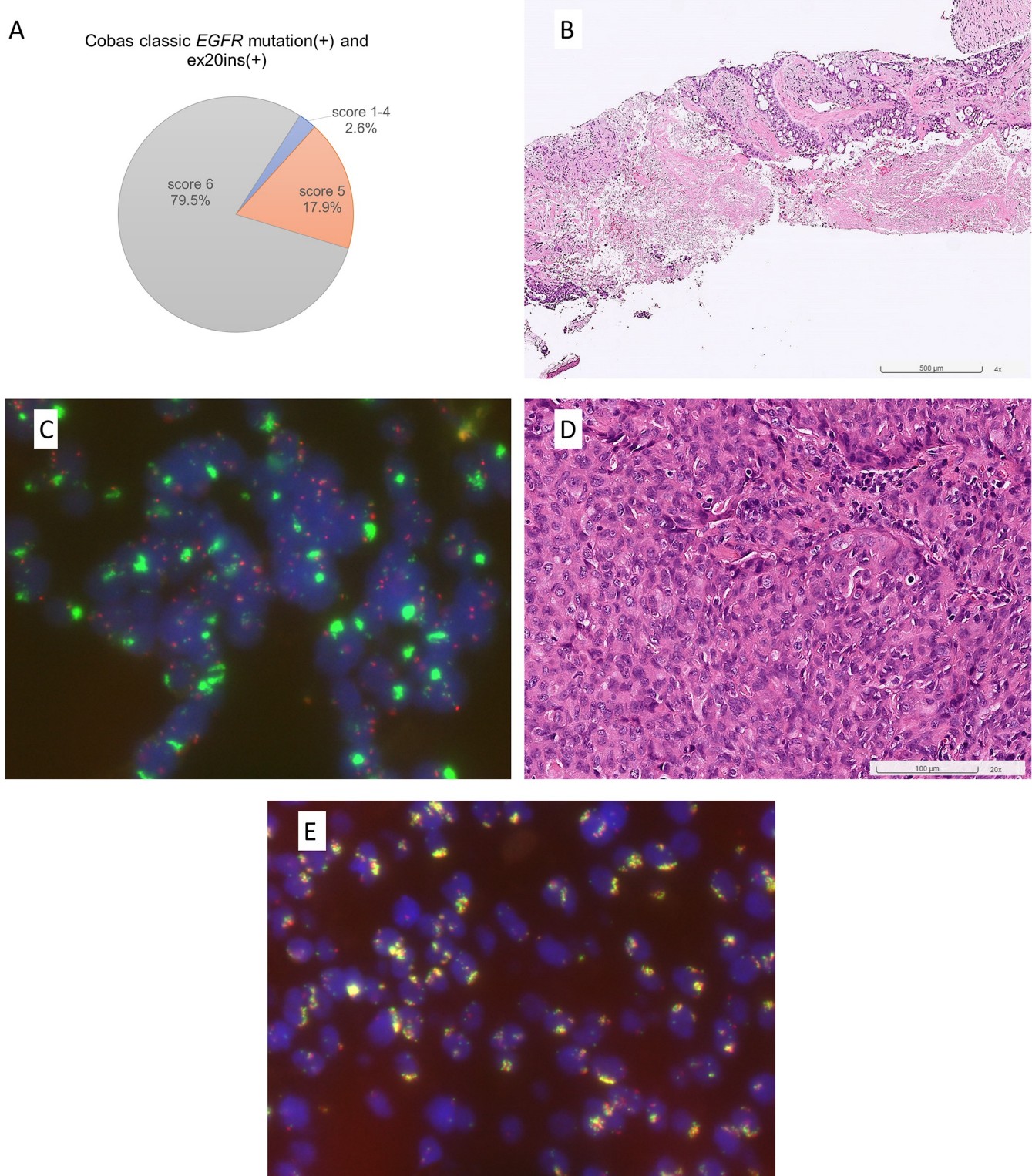

**Fig 4. Results of *EGFR* FISH in cases reported having concomitant classic *EGFR* and ex20ins mutations by the cobas test.** (A) Almost all cases reported having concomitant classic *EGFR* and ex20ins mutations had *EGFR* high polysomy (Colorado score 5, 17.9%) or amplification (Colorado score 6, 79.5%). (B) This case reported as having concomitant classic *EGFR* (ex19del) and ex20ins mutations by cobas testing shows high-grade cribriform patterns with tumor necrosis and (C) *EGFR* amplification (large clusters of *EGFR* signals). (D) This case reported having concomitant classic *EGFR* (L858R) and ex20ins mutations by cobas testing shows high-grade solid patterns with (E) *EGFR* amplification (co-localized clusters of *EGFR* and CEP7 signals). Hematoxylin and eosin (B, D) and *EGFR* FISH, green (*EGFR*) and red (CEN7) signals (C, E).

**Table 5. Comparison of cobas testing, Sanger sequencing, *EGFR* FISH, and liquid NGS results of 20 ex20ins-positive patients reported by the cobas test.**

| Case | cobas test | Sanger sequencing | *EGFR* FISH (Colorado score) | Liquid NGS results | Liquid NGS test |
|---|---|---|---|---|---|
| 1 | ex20ins | p.D770_N771insSVD | No amp (3) | p.D770_N771insSVD | Guardant360 |
| 2 | ex20ins | p.D770_N771insSVD | N/A | p.D770_N771insSVD | FoundationOne Liquid |
| 3 | ex20ins | p.V769_D770insASV | CNG (5) | p.V769_D770insASV | FoundationOne Liquid |
| 4 | ex20ins | p.V769_D770insASV | CNG (5) | p.V769_D770insASV | Guardant360 |
| 5 | ex20ins | p.V769_D770insASV | CNG (5) | p.V769_D770insASV | FoundationOne Liquid |
| 6 | ex20ins | p.V769_D770insASV | CNG (5) | p.V769_D770insASV | Guardant360 |
| 7 | ex20ins | p.V769_D770insASV | No amp (3) | p.V769_D770insASV | FoundationOne Liquid |
| 8 | ex20ins | p.V769_D770insASV | No amp (4) | WT | FoundationOne Liquid |
| 9 | ex20ins | p.H773_V774insH | *EGFR* amp+ (6) | WT | FoundationOne Liquid |
| 10 | ex20ins | WT | *EGFR* amp+ (6) | p.V769_D770insASV | Guardant360 |
| 11 | ex20ins | WT | *EGFR* amp+ (6) | WT | FoundationOne Liquid |
| 12 | ex20ins | WT | *EGFR* amp+ (6) | WT | FoundationOne Liquid |
| 13 | ex20ins | WT | N/A | *EGFR* amp+ | FoundationOne Liquid |
| 14 | ex20ins | WT | *EGFR* amp+ (6) | *EGFR* amp+ | Guardant360 |
| 15 | ex20ins/ex19del | ex19del | *EGFR* amp+ (6) | ex19del, *EGFR* amp+ | FoundationOne Liquid |
| 16 | ex20ins/ex19del | ex19del | No amp (4) | ex19del | FoundationOne Liquid |
| 17 | ex20ins/L858R | L858R | *EGFR* amp+ (6) | L858R, *EGFR* amp+ | FoundationOne Liquid |
| 18 | ex20ins/L858R | L858R | *EGFR* amp+ (6) | L858R, *EGFR* amp+ | FoundationOne Liquid |
| 19 | ex20ins/L858R | L858R | *EGFR* amp+ (6) | L858R, *EGFR* amp+ | FoundationOne Liquid |
| 20 | ex20ins/L858R/T790M | L858R/T790M | *EGFR* amp+ (6) | L858R/T790M | ACT Genomics |

Abbreviations: amp, amplification; CNG, copy number gain; *EGFR*, epidermal growth factor receptor; ex20ins, exon 20 insertion; FISH, fluorescence in situ hybridization; N/A, not applicable; NGS, next-generation sequencing; WT, wild type

astonishingly high *EGFR* amplification rate suggests that the cobas test may yield an aberrant ex20ins report when specimens have *EGFR* amplification.

The distinct ex20ins mutation results from the two different qPCR platforms were most obvious in cases reported to have concomitant classic *EGFR* mut+/ex20ins+. For this group, the agreement between Sanger sequencing and cobas testing was 0% (0/39), but the agreement was almost identical between Sanger sequencing and the Idylla assay (97%, 32/33). This group also had the highest proportion of *EGFR* amplifications (31/39, 79.5%). Cobas and Idylla assays are qPCR platforms that use non-mutated *EGFR* exons as IC. When specimens with similar tumor content were tested, cases with *EGFR* amplification had lower IC Ct values than those without *EGFR* amplification. Unlike the cobas test, the IC Ct values of the Idylla assay were available. The finding that the IC Ct values were significantly lower in the cobas+/Sanger −ex20ins and cobas-identified *EGFR* mut+/ex20ins+ groups is consistent with the high frequencies of *EGFR* amplification in these cases. We believe that the different PCR primers and cutoff thresholds used by these two qPCR platforms led to the dissimilar results. Another example is the p.L858R+K860I doublet mutation [33]. In our department, the p.L858R mutation in cases with the p.L858R+K860I doublet mutation could not be detected by cobas testing but was successfully reported by the Idylla assay. As the primers and cutoff threshold used in the cobas and Idylla platforms are not publicly available, the difference in the design of PCR primers is a reasonable explanation for the inconsistent results for ex20ins and p.L858R +K860I doublet mutations.

*EGFR* ex20ins mutations are typically mutually exclusive from other major driver mutations (including classic *EGFR* mutations such as ex19del, L858R, *ERBB2*, *ALK*, *ROS1*, *BRAF*, or *RET*) but co-occur often with *TP53* mutations and *EGFR* amplification [15, 16,

34, 35]. Studies using NGS found a very low incidence rate of compound classic *EGFR* mut +/ex20ins+ [15, 16]. Qin et al. reported seven cases with concurrent classic *EGFR* mut +/ex20ins+ (7/547, 1.3%) [15]. Riess et al. reported seven cases of concurrent classic *EGFR* mut+/ex20ins+ among 268 ex20ins-positive cases (7/268, 2.6%) [16]. Naidoo et al. reported 46 ex20ins-positive cases but none had concurrent classic *EGFR* mutations as determined by mass spectrometry genotyping [36]. In our hospital, the cobas test reported an unusually high proportion of this co-mutation in ex20ins-positive cases (30/113, 26.5%); however, none could be validated by Sanger sequencing. Cardona et al. reported a very high rate of concurrent *EGFR* mut+/ex20ins+ (36.4% with del19/L858R and 8% with p.G719X/ L861Q/S768I) using the Geno1.2-CLICaP Platform in patients from six Latin American countries (Argentina, Colombia, Costa Rica, Ecuador, Panama, and Mexico) [37]. This extremely high rate of concurrent classic *EGFR* mut+/ex20ins+ among Hispanics requires further study.

In this study, among the Sanger sequencing-confirmed ex20ins-positive cases, 11.3% (8/ 71) showed *EGFR* amplification (*EGFR* FISH Colorado score 6). Previous studies have reported *EGFR* amplification in 17–38.7% of patients with ex20ins mutations [15, 16, 34– 37]. However, the clinical effect of concurrent *EGFR* amplification in patients with ex20ins mutations has rarely been reported. Based on a small number of cases, Gao et al. reported no differences in clinical characteristics, median overall survival, or progression-free survival (in patients treated with chemotherapy) between ex20ins-positive groups with and without *EGFR* amplification [34]. There are also debates regarding the effects of *EGFR* amplification in patients with classic *EGFR* mutations. *EGFR* amplification is more commonly associated with a solid histology pattern, advanced clinical stage, and poorer disease-free survival and is considered a drug resistance mechanism [38]. Sholl et al. observed heterogeneous distribution of *EGFR* amplification mostly in areas with a solid pattern [38]. Our study also showed that *EGFR* amplification was significantly associated with high-grade histology, tumor necrosis, and high tumor content (See S3 Table in S1 File). *EGFR* amplification has been reported as a putative resistance mechanism to EGFR-TKIs in patients with leptomeningeal metastases [39]. However, one study demonstrated that *EGFR* sensitizing mutations and amplification were associated with better overall survival and progression-free survival in Hispanic patients treated with erlotinib [40].

A major limitation of this study was the lack of exact Ct values for the cobas test. The IC used in the cobas test is *EGFR* exon 28, and *EGFR* amplification can lead to a very low IC Ct value. Furthermore, the differences ($\Delta$Ct) between the Ct values of ex20ins mutations and IC are also important. The lower the $\Delta$Ct value, the higher the confidence of the ex20ins result. We believe that the $\Delta$Ct values in the cobas+/Sanger+ group should be much lower than those in the cobas+/Sanger– group. The IC Ct value of the Idylla assay is available, and an IC Ct value of no more than 18 strongly suggests *EGFR* amplification. In addition, the EGFR mutation rate in our department is 71.8% between January 2015 and December 2022, which is higher than the rate reported in previous Taiwanese studies which ranged from 55.5~55.7% [2, 3]. Since our hospital is the largest cancer center in Taiwan and conducting multiple clinical trials of lung cancer, this high EGFR mutation rate may be due to referred cases from other hospitals after diagnosis or disease progression.

In conclusion, the ex20ins mutation reported by the cobas test should be validated by other tests (such as the Idylla assay or NGS), particularly in cases with concomitant classic *EGFR* mut+/ex20ins+. Based on the low IC Ct values and *EGFR* FISH results, *EGFR* amplification may have led to aberrant ex20ins results in the cobas report.

## Supporting information

**S1 File. S1-S3 Tables.**
(DOCX)

**S2 File. Clinicopathological features and results of EGFR testing of all cases.**
(XLSX)

## Acknowledgments

We are grateful to the staff of the Department of Pathology, National Taiwan University Hospital for their support.

## Author Contributions

**Conceptualization:** Man-San Zhang, Min-Shu Hsieh.

**Funding acquisition:** Min-Shu Hsieh.

**Investigation:** Man-San Zhang, Lai-Jin Yao, Min-Shu Hsieh.

**Methodology:** Man-San Zhang, Min-Shu Hsieh.

**Supervision:** Yi-Chen Yeh, Hsien-Neng Huang, Long-Wei Lin, Yen-Lin Huang, Lei-Chi Wang, Lai-Jin Yao, Tze-Chun Hung, Yu-Fen Tseng, Yi-Hsuan Lee, Wei-Yu Liao.

**Writing – original draft:** Man-San Zhang, Min-Shu Hsieh.

**Writing – review & editing:** Wei-Yu Liao, Jin-Yuan Shih, Min-Shu Hsieh.

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
