## [Decision Letter · Decision Letter 0]

23 Nov 2023

PONE-D-23-26697The association of EGFR amplification with aberrant exon 20 insertion report using the cobas EGFR Mutation Test v2PLOS ONE

Dear Dr. Hsieh,

Thank you for submitting your manuscript to PLOS ONE. After careful consideration, we feel that it has merit but does not fully meet PLOS ONE’s publication criteria as it currently stands. Therefore, we invite you to submit a revised version of the manuscript that addresses the points raised during the review process.

We look forward to receiving your revised manuscript.

Kind regards,

Fumihiro Yamaguchi

Academic Editor

PLOS ONE

Journal Requirements:

Reviewers' comments:

Reviewer's Responses to Questions

**Comments to the Author**

1. Is the manuscript technically sound, and do the data support the conclusions?

Reviewer #1: Partly

Reviewer #2: Partly

2. Has the statistical analysis been performed appropriately and rigorously? 

Reviewer #1: Yes

Reviewer #2: No

3. Have the authors made all data underlying the findings in their manuscript fully available?

Reviewer #1: No

Reviewer #2: Yes

4. Is the manuscript presented in an intelligible fashion and written in standard English?

Reviewer #1: Yes

Reviewer #2: Yes

5. Review Comments to the Author

Reviewer #1: I appreciate the opportunity to provide a peer-review of such an interesting article. The authors present a scenario that is rarely contemplated in our everyday clinical practice, which is to evaluate whether the results of a test to determine EGFR mutations, previously validated by regulatory agencies, could lead to aberrant results. The authors noticed that the introduction of the new technique (Cobas) resulted in a higher frequency of co-occurring mutations (considered very rare), including both typical EGFR mutations and the relatively uncommon ex20ins mutations, which are traditionally considered mutually exclusive. Subsequently, they re-examined the samples using alternative methods, including Sanger, Idylla, and even NGS, only to find that none of these techniques confirmed the presence of ex20ins mutations. Moreover, the authors noted that in cases where Cobas detected ex20ins mutations while Sanger did not, more aggressive disease characteristics were observed, which instigated investigations into EGFR amplification. The finding is interesting as it suggests the possibility that EGFR amplification might be a contributing factor to the aberrant results reported by the Cobas technique. Finally, the authors conclude with the recommendation to validate ex20ins results (whit another test) when reported by the Cobas technique, and that EGFR amplification can lead to aberrant ex20ins mutation results. Given that recent data on the benefit of poziotinib and amivantamab in patients with ex20ins have been reported, this recommendation is particularly relevant. Additionally, given the regulatory agencies' endorsement of the Cobas test, any article that calls its accuracy into question should provide a comprehensive and meticulously structured body of evidence to support the hypotheses of aberrant outcomes. That said, I have some major concerns along with some minor ones:

Major Comments:

1. In the third paragraph of the introduction, some procedures performed in the study are described, and results of these studies are also described. This should be specified in detail in the methods and results sections, respectively. I suggest removing this information from the introduction section. On the other hand, within the introduction, I suggest including statements about the Cobas and Sanger techniques since the entire article is about the accuracy of these tests, so it is important to have a paragraph with data about the accuracy of these platforms and what is known about these tests.

2. In the final part of the introduction, the authors state, "... The study aimed to COMPARE the clinical-pathological features and EGFR amplification status ...", however, the study's conclusions indicate that the ex20ins test reported by Cobas should be validated by other techniques and that EGFR amplification can lead to aberrant results. The study's conclusions do not align with the study's objective. I kindly suggest that the authors describe the study's objective in a way that the conclusions correspond to that objective.

3. Within the methods section, the authors do not provide details about the study's design. Based on the study's features and the data gathered, it appears to be a cross-sectional, descriptive study. I would recommend that they clarify the study's design, because the results are hypothesis generating, and this should be taken into account when analyzing results.

4. The article could be structured in a logical sequence that starts with the observation of concurrent mutations => evaluation of ex20ins by other techniques => observation of result discordance => clinical-pathological evaluation and FISH of the samples => Description of aberrant results and the hypothesis that amplification could explain the discordant results. I kindly suggest to the authors to consider restructuring the article in this way

5. I recommend revisiting the title and adjusting the abstract to align with the previous modifications.

Minor Comments:

- In the results (Table 01), the high frequency of EGFR mutations found in the reviewed cases (71%) is noteworthy, which could be associated with selection bias, the descriptive design and the retrospective data collection. I kindly suggest to the authors commenting on these findings

- In the third line of the abstract, "ETFT" was written. I suggest correcting this typo

- In Table 01, correct the typo:

- In the "near-loop" row, the last column should have an open parenthesis

- In the "far-loop" row, the last column should have an open parenthesis.

Best Regards

Reviewer #2: Please could you elaborate on this statement: "In Taiwan, epidermal growth factor receptor (EGFR) mutations are the most common disease-associated mutations in lung cancer, and several EGFR tyrosine kinase inhibitors (EGFR-TKIs) are used as first-line therapy [4-8]" What does the literature say about the high incidence of lung cancer among Taiwanese people?

I would appreciate it if you could provide more details regarding why patients with uncommon EGFR mutations, such as p.G719X, p.S768I, or p.L861Q, typically display limited response to EGFR-TKIs.?

The introduction requires significant modifications to imbue a scientific tone.

The author utilized one-way ANOVA to conduct a comparison between the three groups. I'm interested in knowing the specific criteria considered when choosing Colorado 1 to 4 as the main focus for the analysis. Furthermore, did the author assess the normality of the data prior to conducting this test?

6. PLOS authors have the option to publish the peer review history of their article (what does this mean?). If published, this will include your full peer review and any attached files.

Reviewer #1: No

Reviewer #2: **Yes: **ZIYAD F AL NUFAIEI

---

## [Author Response · Author response to Decision Letter 0]

8 Dec 2023

Response to Reviewers

Dear editor and reviewers, 

We all are grateful for your comments and revised our manuscript according to your suggestions. 

Reviewer #1: I appreciate the opportunity to provide a peer-review of such an interesting article. The authors present a scenario that is rarely contemplated in our everyday clinical practice, which is to evaluate whether the results of a test to determine EGFR mutations, previously validated by regulatory agencies, could lead to aberrant results. The authors noticed that the introduction of the new technique (Cobas) resulted in a higher frequency of co-occurring mutations (considered very rare), including both typical EGFR mutations and the relatively uncommon ex20ins mutations, which are traditionally considered mutually exclusive. Subsequently, they re-examined the samples using alternative methods, including Sanger, Idylla, and even NGS, only to find that none of these techniques confirmed the presence of ex20ins mutations. Moreover, the authors noted that in cases where Cobas detected ex20ins mutations while Sanger did not, more aggressive disease characteristics were observed, which instigated investigations into EGFR amplification. The finding is interesting as it suggests the possibility that EGFR amplification might be a contributing factor to the aberrant results reported by the Cobas technique. Finally, the authors conclude with the recommendation to validate ex20ins results (whit another test) when reported by the Cobas technique, and that EGFR amplification can lead to aberrant ex20ins mutation results. Given that recent data on the benefit of poziotinib and amivantamab in patients with ex20ins have been reported, this recommendation is particularly relevant. Additionally, given the regulatory agencies' endorsement of the Cobas test, any article that calls its accuracy into question should provide a comprehensive and meticulously structured body of evidence to support the hypotheses of aberrant outcomes. That said, I have some major concerns along with some minor ones:

Major Comments:

1. In the third paragraph of the introduction, some procedures performed in the study are described, and results of these studies are also described. This should be specified in detail in the methods and results sections, respectively. I suggest removing this information from the introduction section. On the other hand, within the introduction, I suggest including statements about the Cobas and Sanger techniques since the entire article is about the accuracy of these tests, so it is important to have a paragraph with data about the accuracy of these platforms and what is known about these tests.

Response to Reviewer:

Thank you for your suggestion. We've added information regarding the accuracy of these tests after the second paragraph of our introduction:

「Several molecular testing methods has seen used to detect druggable EGFR mutations. Sanger sequencing has traditionally been considered as the gold standard method but with a low sensitivity which typically requires 20% of mutated alleles [23]. Real-time polymerase chain reaction (RT-PCR)-based tests have been developed and widely used in Taiwan. The cobas EGFR Mutation Test v2 (Roche Molecular Diagnostics, Pleasanton, CA, USA) is an FDA approved RT-PCR test detecting 42 specific mutations in exons 18 to 21 of the human EGFR gene, and it is more sensitive than Sanger sequencing with a detection limit of ~5% of mutant alleles. The IdyllaTM EGFR Mutation Test (Biocartis, Mechelen, Belgium) is a fully automated RT-PCR based molecular diagnostic kit for the qualitative detection of 51 EGFR mutations for FFPE tissue samples with a limit of detection ~2-5% of variant allele frequency [24,25]. Next-generation sequencing (NGS) is more sensitive than RT-PCR based tests with a limit of detection around 1% variant allele frequency [26].」

2. In the final part of the introduction, the authors state, "... The study aimed to COMPARE the clinical-pathological features and EGFR amplification status ...", however, the study's conclusions indicate that the ex20ins test reported by Cobas should be validated by other techniques and that EGFR amplification can lead to aberrant results. The study's conclusions do not align with the study's objective. I kindly suggest that the authors describe the study's objective in a way that the conclusions correspond to that objective.

Response to Reviewer:

Thank you for your thoughtful consideration of our study and and providing valuable insights about the match between our objective and conclusions. Based on your suggestion, we'll adjust the study's objective in the final part of the introduction to better match our research path and align with the conclusions:

「In summary, we reported cases reported to have EGFR ex20ins mutation by the cobas test but unable to be validated by Sanger sequencing, Idylla assay, or liquid biopsy. Comparing with ex20ins cobas+/Sanger+ cases, these ex20ins cobas+/Sanger- cases groups were significantly associated with higher tumor contents, high-grade patterns, tumor necrosis, lower IC Ct value, and EGFR amplification. Based on the low IC Ct values and EGFR FISH results, EGFR amplification may have led to aberrant ex20ins results in the cobas report. It is recommended that ex20ins mutation reported by the cobas test should be validated by other tests (such as the Idylla assay or NGS), particularly in cases with concomitant classic EGFR mut+/ex20ins+.」

3. Within the methods section, the authors do not provide details about the study's design. Based on the study's features and the data gathered, it appears to be a cross-sectional, descriptive study. I would recommend that they clarify the study's design, because the results are hypothesis generating, and this should be taken into account when analyzing results.

Response to Reviewer:

Thank you for your comment. We added a brief description of our study design in the last part of our instruction:

「This study aimed to validate the ex20ins results reported by the cobas test and to determine whether the different testing sensitivities between cobas and Sanger sequencing leading to the disparity of ex20ins results. Clinicopathological features including tumor contents were compared between cobas+/Sanger+ and cobas+/Sanger- ex20ins groups, and the ex20ins results were validated by Idyalla EGFR Assay. The observation that Sanger sequencing had higher agreement with Idylla and the lower IC Ct value in the cobas+/Sanger- ex20ins group raised the suspicion that high EGFR copy numbers may be associated with aberrant cobas ex20ins report. Fluorescence in situ hybridization (FISH) was used to determine whether there were differences in EGFR gene copy numbers like amplification between these two groups. Finally, medical records including liquid NGS tests and clinical response to EGFR TKIs were reviewed.」

We also modified our methods section: 

In the first part “Case selection and design of the study”, we added 

“This study was a cross-sectional study. The first aim was to compare tumor contents between cobas+/Sanger+ and cobas+/Sanger- ex20ins groups to determine whether the different testing sensitivities of cobas and Sanger sequencing leading to the disparity of ex20ins results. Clinicopathological features including tumor contents were compared between cobas+/Sanger+ and cobas+/Sanger- ex20ins groups. Then Idyalla EGFR Assay was used to validate the ex20ins results of both groups, and the IC Ct values from two groups were compared.” 

In the second part “Mutation analysis”, we added “IC Ct values between ex20ins cobas+/Sanger+ and cobas+/Sanger- groups.were compared.” after the description of Idylla EGFR Mutation Testing.

In the third part “EGFR FISH”, we added “ FISH was used to compare EGFR gene copy numbers between x20ins cobas+/Sanger+ and cobas+/Sanger- groups using the Colorado Scoring Criteria. Fig 1 shows the algorithm for groups cases for EGFR FISH..”

4. The article could be structured in a logical sequence that starts with the observation of concurrent mutations => evaluation of ex20ins by other techniques => observation of result discordance => clinical-pathological evaluation and FISH of the samples => Description of aberrant results and the hypothesis that amplification could explain the discordant results. I kindly suggest to the authors to consider restructuring the article in this way

5. I recommend revisiting the title and adjusting the abstract to align with the previous modifications.

Response to Reviewer:

We are grateful for your kind suggestion which makes our manuscript more readable. We rearranged our results according to your recommendation as:

[1] The emergence of cases reported to have concomitant ex20ins and classic EGFR mutations after the use of the cobas test and discrepancies with Sanger sequencing

[2] The cobas+/Sanger- ex20ins group had a higher tumor content, a higher histologic grade, more frequent tumor necrosis, and a lower IC Ct value

[3] Significant disparity in rates of EGFR amplification between cobas+/Sanger- and cobas+/Sanger+ groups

[4] The majority of cases reported having concomitant classic EGFR mut+/ex20ins+ by the cobas test had EGFR amplification

[5] Comparison of cobas testing, Sanger sequencing, EGFR FISH, and liquid NGS results.

All authors agreed that our manuscript became more organized after this modification. We all are very grateful for your kind suggestion.

Minor Comments:

- In the results (Table 01), the high frequency of EGFR mutations found in the reviewed cases (71%) is noteworthy, which could be associated with selection bias, the descriptive design and the retrospective data collection. I kindly suggest to the authors commenting on these findings

- In the third line of the abstract, "ETFT" was written. I suggest correcting this typo

- In Table 01, correct the typo:

- In the "near-loop" row, the last column should have an open parenthesis

- In the "far-loop" row, the last column should have an open parenthesis.

Response to Reviewer:

Thank you so much for your comment. We added 「In addition, the EGFR mutation rate in our department is 71.8% between January 2015 and December 2022, which is higher than the rate reported in previous Taiwanese studies which ranged from 55.5~55.7% [6, 39]. Since our hospital is the largest cancer center in Taiwan and conducting multiple clinical trials of lung cancer, this high EGFR mutation rate may be due to referred cases from other hospitals after diagnosis or disease progression.」in our discussion.

We also corrected errors of “ETFT” in our abstract and the Table 1.

 

Reviewer #2: 

1. Please could you elaborate on this statement: "In Taiwan, epidermal growth factor receptor (EGFR) mutations are the most common disease-associated mutations in lung cancer, and several EGFR tyrosine kinase inhibitors (EGFR-TKIs) are used as first-line therapy [4-8]" What does the literature say about the high incidence of lung cancer among Taiwanese people?

Response to Reviewer:

Thank you for raising this point. It has been shown that the EGFR mutation rate is much higher in Asia (~40-50%, including Taiwan, Japan, Korea, and China) than that in the Europe or USA (~15%). According to previous studies in Taiwan, the EGFR mutation rate is around 55%. 

Ref 4: Wu JY, Wu SG, Yang CH, et al. Lung cancer with epidermal growth factor receptor exon 20 mutations is associated with poor gefitinib treatment response. Clin Cancer Res. 2008;14:4877-4882. https://doi.org/10.1158/1078-0432.ccr-07-5123

Ref 5: KH Hsu, CC Ho, TC Hsia, et al. Identification of Five Driver Gene Mutations in Patients with Treatment-Naïve Lung Adenocarcinoma in Taiwan. PLoS ONE. 2015;10: e0120852

We revised our description as: 

「In Taiwan, epidermal growth factor receptor (EGFR) mutations are the most common disease-associated mutations in lung cancer with an incidence rate around 55% [4, 5], and several EGFR tyrosine kinase inhibitors (EGFR-TKIs) are used as first-line therapy [4-9].」

2. I would appreciate it if you could provide more details regarding why patients with uncommon EGFR mutations, such as p.G719X, p.S768I, or p.L861Q, typically display limited response to EGFR-TKIs.?

Response to Reviewer:

Thank you for your inquiry regarding the limited response observed in patients with uncommon EGFR mutations, including p.G719X, p.S768I, and p.L861Q, to EGFR tyrosine kinase inhibitors (EGFR-TKIs). 

Robichaux et al (Nature. 2021 Sep;597(7878):732-737) established the structure-function relationship of EGFR mutations on drug sensitivity and categorized EGFR mutations into four distinct subgroups on the basis of sensitivity and structural changes that retrospectively predict patient outcomes following treatment. These four subgroups are classical-like, T790M-like, ex20 loop insertion, and P-loop αC-helix compression types. Uncommon mutations like G719X, L861Q, and S768I represent mutations proximal to drug binding pocket which has direct or indirect impact on drug binding which make first generation EGFR TKIs less effective.

We have added more detailed information in the second paragraph of the article introduction:

「Based on the structural changes and drug sensitivity, EGFR mutations can be separated into four distinct subgroups as classical-like, T790M-like, ex20 loop insertion, and P-loop αC-helix compression types [11]. Uncommon mutations like G719X, L861Q, and S768I represent mutations proximal to drug binding pocket which has direct or indirect impact on drug binding which make first generation EGFR TKIs less effective [8, 11]. Afatinib, a second generation EGFR TKI, has been demonstrated to be more effective for NSCLCs carrying these uncommon mutations and now considered as the first-line therapy with an overall response rate (ORR) exceeding 50% and a median time to treatment failure (TTF) of nearly one year [9, 12].」

3. The introduction requires significant modifications to imbue a scientific tone. 

Response to Reviewer:

Thank you for your comment. We added a brief description of our study design in the last part of our instruction to delineate our study design and aims.

「This study aimed to validate the ex20ins results reported by the cobas test and to determine whether the different testing sensitivities between cobas and Sanger sequencing leading to the disparity of ex20ins results. Clinicopathological features including tumor contents were compared between cobas+/Sanger+ and cobas+/Sanger- ex20ins groups, and the ex20ins results were validated by Idyalla EGFR Assay. The observation that Sanger sequencing had higher agreement with Idylla and the lower IC Ct value in the cobas+/Sanger- ex20ins group raised the suspicion that high EGFR copy numbers may be associated with aberrant cobas ex20ins report. Fluorescence in situ hybridization (FISH) was used to determine whether there were differences in EGFR gene copy numbers like amplification between these two groups. Finally, medical records including liquid NGS tests and clinical response to EGFR TKIs were reviewed.」

4. The author utilized one-way ANOVA to conduct a comparison between the three groups. I'm interested in knowing the specific criteria considered when choosing Colorado 1 to 4 as the main focus for the analysis. Furthermore, did the author assess the normality of the data prior to conducting this test?

Response to Reviewer:

Thank you for your comment. The Colorado score has been established and modified by M Varella-Garcia. According to his guidelines for application EGFR FISH to non-small-cell lung and multiple clinical trials, the cut-off value for EGFR FISH positivity was high polysomy (Colorado score 5, defined as ≥4 copies of EGFR in >40% of cells) or EGFR gene amplification (Colorado score=6, defined as (a) an EGFR to CEP7 ratio ≥2 over all scored nuclei and calculated using the sum of EGFR divided by the sum of CEP7 when mean CEP7 per cell is ≥2 copies, or (b) the presence of gene cluster (≥4 spots) in ≥10% of tumor cells, or (c) at least 15 copies of the EGFR signals in ≥10% of tumor cells.). Cases with Colorado score 5 or 6 are considered as EGFR FISH-positive, while those with score 1-4 are considered EGFR FISH-negative according to the guideline [Ref 28]. The background non-tumor cells like inflammatory cells served as normal control for FISH study just like other FISH tests performed in our hospital.

Ref 28. Varella-Garcia M, Diebold J, Eberhard DA, et al. EGFR fluorescence in situ hybridisation assay: guidelines for application to non-small-cell lung cancer. J Clin Pathol. 2009;62:970-977. doi: 10.1136/jcp.2009.066548.

---

## [Decision Letter · Decision Letter 1]

10 Jan 2024

PONE-D-23-26697R1The association of EGFR amplification with aberrant exon 20 insertion report using the cobas EGFR Mutation Test v2PLOS ONE

Dear Dr. Hsieh,

Thank you for submitting your manuscript to PLOS ONE. After careful consideration, we feel that it has merit but does not fully meet PLOS ONE’s publication criteria as it currently stands. Therefore, we invite you to submit a revised version of the manuscript that addresses the points raised during the review process.

**ACADEMIC EDITOR: **While your paper is interesting, there is still room for improvement. It would be an even better paper if you could address the issues raised by the reviewers.

We look forward to receiving your revised manuscript.

Kind regards,

Fumihiro Yamaguchi

Academic Editor

PLOS ONE

Reviewers' comments:

Reviewer's Responses to Questions

**Comments to the Author**

1. If the authors have adequately addressed your comments raised in a previous round of review and you feel that this manuscript is now acceptable for publication, you may indicate that here to bypass the “Comments to the Author” section, enter your conflict of interest statement in the “Confidential to Editor” section, and submit your "Accept" recommendation.

Reviewer #1: (No Response)

Reviewer #2: All comments have been addressed

2. Is the manuscript technically sound, and do the data support the conclusions?

Reviewer #1: Partly

Reviewer #2: Yes

3. Has the statistical analysis been performed appropriately and rigorously? 

Reviewer #1: Yes

Reviewer #2: Yes

4. Have the authors made all data underlying the findings in their manuscript fully available?

Reviewer #1: No

Reviewer #2: Yes

5. Is the manuscript presented in an intelligible fashion and written in standard English?

Reviewer #1: No

Reviewer #2: Yes

6. Review Comments to the Author

Reviewer #1: Dear authors:

I appreciate the willingness and openness to suggestions and comments, as well as your review of the original manuscript. I believe that the findings you wish to convey are particularly relevant to daily clinical practice, as they provide crucial information for decision-making in the management of patients with advanced lung cancer with EGFR mutation. I think you have done a great job in this study. I understand that summarizing and "simplifying" the findings is challenging given the complexity of the results. However, the way you communicate your results is as important as the findings themselves, so presenting them in a scientific and organized manner should be the ultimate goal of the manuscript. Therefore, I still have some significant observations about your manuscript:

* In the introduction section, established knowledge relevant to the study should be included, along with the observations or knowledge gaps that led to conducting the study, without repeating the described information. Therefore, I kindly suggest:

- Consider the possibility of moving the sentence referring to EGFR mutation subgroups that states, “…Based on the structural…” (lines 15 to 18 on page 5) to the end of the first paragraph (after “…afatinib, and osimertinib [4-7]”), and omitting the sentence 'Uncommon mutation like…TKIs less effective [8,11]' (lines 17 to 19, page 5) because it is redundant information already included in the final sentence of the first paragraph. (It would have been helpful to have the draft with line numbers to explain this more effectively).

- On page 6, line 9, from the sentence 'Cell line studies...' to '...resistant to conventional EGFR-TKI...' repeats information already described in line 6 of the same page.

- In the penultimate line of page 6, it is stated as a fact that the COBAS method is more sensitive than Sanger. However, since the entire study focuses on the possibility of aberrant findings with the COBAS method, I suggest that the authors consider presenting it not as an established fact but rather as follows: 'Greater sensitivity of the COBAS method has been described.' Similarly, in the paragraph starting on line 10 of page 7.

- From line 14 of page 7 ('...Furthermore, cases reported...') to line 5 of page 8, findings from the study are described that should be included in the results section.

- I kindly suggest reviewing the second sentence in the introduction section regarding NCCN recommendations (lines 2 to 5, page 5) and the section about Afatinib (line 19, page 5, to line 1, page 6) to reconsider if the information described there is relevant to the manuscript. My suggestion is to remove them.

- In line 16, '… has seen…' is used. I suggest reviewing the typo.

* In the Methods section, I kindly suggest to the authors to organize the included information by first describing the study to be conducted (observational, descriptive, cross-sectional design) based on the observations, and then detailing the selection of cases to be reviewed.

* In the Discussion section, the interpretation of the results obtained in the current context and their practical application should be included. The first two paragraphs of the discussion (up to line 12 on page 23) provide information about the EGFR exon 20 insertion mutation and tests for determining EGFR, which should be summarized in the introduction section. I kindly suggest to the authors to review and condense this information together in the introduction, being careful not to repeat the information.

* I kindly suggest the authors to rewrite the abstract according to the previous changes.

Best Regards

Reviewer #2: Thank you for your time and effort. The author worked hard to address all the comments I have suggested.

7. PLOS authors have the option to publish the peer review history of their article (what does this mean?). If published, this will include your full peer review and any attached files.

Reviewer #1: No

Reviewer #2: **Yes: **ZIYAD F AL NUFAIEI

---

## [Author Response · Author response to Decision Letter 1]

25 Jan 2024

Response to Reviewers

Reviewer #1: Dear authors:

I appreciate the willingness and openness to suggestions and comments, as well as your review of the original manuscript. I believe that the findings you wish to convey are particularly relevant to daily clinical practice, as they provide crucial information for decision-making in the management of patients with advanced lung cancer with EGFR mutation. I think you have done a great job in this study. I understand that summarizing and "simplifying" the findings is challenging given the complexity of the results. However, the way you communicate your results is as important as the findings themselves, so presenting them in a scientific and organized manner should be the ultimate goal of the manuscript. Therefore, I still have some significant observations about your manuscript:

Response to Reviewer: 

All authors deeply appreciate your review and kind suggestions. We are grateful to have the chance to revise our manuscript. We also uploaded all detailed information of this study in the file “Supplementary information_all cases”.

* In the introduction section, established knowledge relevant to the study should be included, along with the observations or knowledge gaps that led to conducting the study, without repeating the described information. Therefore, I kindly suggest:

- Consider the possibility of moving the sentence referring to EGFR mutation subgroups that states, “…Based on the structural…” (lines 15 to 18 on page 5) to the end of the first paragraph (after “…afatinib, and osimertinib [4-7]”), and omitting the sentence 'Uncommon mutation like…TKIs less effective [8,11]' (lines 17 to 19, page 5) because it is redundant information already included in the final sentence of the first paragraph. (It would have been helpful to have the draft with line numbers to explain this more effectively).

Response to Reviewer: 

Thank you for your kind comment. We moved the sentence “Based on the structural…” to the end of the first paragraph, and omitted the sentence “Uncommon mutation like… TKIs less effective [8,11]” according to your suggestion.

- On page 6, line 9, from the sentence 'Cell line studies...' to '...resistant to conventional EGFR-TKI...' repeats information already described in line 6 of the same page.

Response to Reviewer: 

Thank you for your comment. We deleted “Cell line studies have demonstrated that exon20ins mutations in the near-loop region (positions 767–772) are resistant to conventional EGFR-TKIs but sensitive to poziotinib, while those in the far-loop region (positions 773–774) are sensitive to second-generation EGFR-TKIs (i.e., afatinib) [11]” according to your suggestion.

- In the penultimate line of page 6, it is stated as a fact that the COBAS method is more sensitive than Sanger. However, since the entire study focuses on the possibility of aberrant findings with the COBAS method, I suggest that the authors consider presenting it not as an established fact but rather as follows: 'Greater sensitivity of the COBAS method has been described.' Similarly, in the paragraph starting on line 10 of page 7.

Response to Reviewer: 

We deleted the description of COBAS more sensitive than Sanger and revised the sentence as “The cobas EGFR Mutation Test v2 (Roche Molecular Diagnostics, Pleasanton, CA, USA) is an FDA approved RT-PCR test detecting 42 specific mutations in exons 18 to 21 of the human EGFR gene, and it is more sensitive than Sanger sequencing with a detection limit of ~5% of mutant alleles.” We also revised the sentence “Although greater sensitivity of the cobas method has been described, the cobas test is more sensitive than Sanger sequencing,ex20ins mutations in many cases (approximately one-third of ex20ins cases reported by cobas) could not be confirmed by Sanger sequencing…” according to your kind suggestion.

- From line 14 of page 7 ('...Furthermore, cases reported...') to line 5 of page 8, findings from the study are described that should be included in the results section.

Response to Reviewer: 

Thank you for your comment. We removed this part from the introduction section. 

- I kindly suggest reviewing the second sentence in the introduction section regarding NCCN recommendations (lines 2 to 5, page 5) and the section about Afatinib (line 19, page 5, to line 1, page 6) to reconsider if the information described there is relevant to the manuscript. My suggestion is to remove them.

Response to Reviewer: 

We removed these parts from the introduction section according to your suggestion.

- In line 16, '… has seen…' is used. I suggest reviewing the typo.

Response to Reviewer: 

Thank you for your comment. We replaced “has seen” with “have been”. 

* In the Methods section, I kindly suggest to the authors to organize the included information by first describing the study to be conducted (observational, descriptive, cross-sectional design) based on the observations, and then detailing the selection of cases to be reviewed.

Response to Reviewer: 

Thank you for your comment. We added “Based on the observation that cobas test might give aberrant EGFR ex20ins reports, a multicenter, cross-sectional study was designed to compare the ex20ins results among different testing methods” before the detailing the selection of cases to be reviewed according to your kind suggestion. 

* In the Discussion section, the interpretation of the results obtained in the current context and their practical application should be included. The first two paragraphs of the discussion (up to line 12 on page 23) provide information about the EGFR exon 20 insertion mutation and tests for determining EGFR, which should be summarized in the introduction section. I kindly suggest to the authors to review and condense this information together in the introduction, being careful not to repeat the information.

Response to Reviewer: 

Thank you for your comment. We summarized and moved the first two paragraphs of the discussion to the introduction section according to your kind suggestion. We also deleted the sentence “The Idylla EGFR assay is a qPCR platform with a limit of detection varying from 2–5% variant allele frequency. It detects 51 EGFR mutations, including exon 18 (p.G719A/S/C), 36 ex19dels, exon 20 (p.T790M, p.S768I), five ex20ins mutations (identical mutation types with the cobas test), and exon 21 point mutations (p.L858R, p.L861Q) [24,25]” in the discussion as this information has been mentioned in the introduction section.

* I kindly suggest the authors to rewrite the abstract according to the previous changes.

Response to Reviewer: 

We are grateful for your kind recommendation and we revised our abstract according to previous changes. 

Best Regards

Reviewer #2: Thank you for your time and effort. The author worked hard to address all the comments I have suggested.

Response to Reviewer:

All authors are grateful for your review and kind suggestions. We are grateful to have the chance to revise our manuscript.

---

## [Decision Letter · Decision Letter 2]

12 Mar 2024

The association of EGFR amplification with aberrant exon 20 insertion report using the cobas EGFR Mutation Test v2

PONE-D-23-26697R2

Dear Dr. Hsieh,

We’re pleased to inform you that your manuscript has been judged scientifically suitable for publication and will be formally accepted for publication once it meets all outstanding technical requirements.

Kind regards,

Fumihiro Yamaguchi

Academic Editor

PLOS ONE

Additional Editor Comments (optional):

Dear Author,

Thank you for a very valuable article. You have responded accurately to all the reviewers' comments.

Reviewers' comments:

Reviewer's Responses to Questions

**Comments to the Author**

1. If the authors have adequately addressed your comments raised in a previous round of review and you feel that this manuscript is now acceptable for publication, you may indicate that here to bypass the “Comments to the Author” section, enter your conflict of interest statement in the “Confidential to Editor” section, and submit your "Accept" recommendation.

Reviewer #1: All comments have been addressed

2. Is the manuscript technically sound, and do the data support the conclusions?

Reviewer #1: Yes

3. Has the statistical analysis been performed appropriately and rigorously? 

Reviewer #1: Yes

4. Have the authors made all data underlying the findings in their manuscript fully available?

Reviewer #1: Yes

5. Is the manuscript presented in an intelligible fashion and written in standard English?

Reviewer #1: Yes

6. Review Comments to the Author

Reviewer #1: (No Response)

7. PLOS authors have the option to publish the peer review history of their article (what does this mean?). If published, this will include your full peer review and any attached files.

Reviewer #1: No

---

## [Editor Report · Acceptance letter]

18 Mar 2024

PONE-D-23-26697R2 

PLOS ONE

Dear Dr. Hsieh, 

I'm pleased to inform you that your manuscript has been deemed suitable for publication in PLOS ONE. Congratulations! Your manuscript is now being handed over to our production team.

Kind regards, 

on behalf of

Dr. Fumihiro Yamaguchi 

Academic Editor

PLOS ONE